# Untangling the seasonal dynamics of plant-pollinator communities

Bernat Bramon Mora[1,2 ✉], Eura Shin[1,3], Paul J. CaraDonna[4,5] & Daniel B. Stouffer [1]

Ecological communities often show changes in populations and their interactions over time. To date, however, it has been challenging to effectively untangle the mechanisms shaping such dynamics. One approach that has yet to be fully explored is to treat the varying structure of empirical communities—i.e. their network of interactions—as time series. Here, we follow this approach by applying a network-comparison technique to study the seasonal dynamics of plant-pollinator networks. We find that the structure of these networks is extremely variable, where species constantly change how they interact with each other within seasons. Most importantly, we find the holistic dynamic of plants and pollinators to be remarkably coherent across years, allowing us to reveal general rules by which species first enter, then change their roles, and finally leave the networks. Overall, our results disentangle key aspects of species' interaction turnover, phenology, and seasonal assembly/disassembly processes in empirical plant-pollinator communities.

[1] Centre for Integrative Ecology, School of Biological Sciences, University of Canterbury, Christchurch, New Zealand. [2] Institute of Integrative Biology, ETH Zürich, Zürich, Switzerland. [3] University of Kentucky, Lexington, KY, USA. [4] Chicago Botanic Garden, Chicago, IL, USA. [5] Rocky Mountain Biological Laboratory, Crested Butte, CO, USA. ✉email: bernat.bramon@gmail.com

Ecological communities are inherently dynamic. Their species composition is in constant change due to species' intrinsic phenologies[1], environmental variability[2,3], and disturbances such as habitat fragmentation and invasive species[4,5]. In turn, the presence, absence, and intensity of ecological interactions also vary over time[6]. This happens either by the direct turnover of interacting species or by higher-order effects of changes in the community composition[7]. That is, the arrival of a new species in a community will come hand-in-hand with a new set of interactions, and these changes in the community will also indirectly interfere with existing interactions (e.g. potentially generating new cases of apparent competition between species[8]).

There is a longstanding tradition in ecology of exploring community dynamics using mathematical models[9,10]. The empirical basis of such models is generally static interaction networks, where nodes and links represent the different species and their observed interactions aggregated over sampling seasons[7,11,12]. While species' interaction strengths are linked to their abundances in these models, the aforementioned dynamic nature of ecological interactions makes the underlying static representation of a community unrealistic—for example, plant-pollinator systems have been shown to present high levels of within-season species and interaction turnover[13,14]. Fortunately, several pioneering empirical studies have laid the groundwork for analysing natural systems over different time scales, providing crucial examples of the way ecological communities change within seasons[1,15–17], across seasons[18,19], and over much longer time scales[20,21]. These examples can often be represented as network time series, providing glimpses of ecological dynamics at whole-community scales. Such a network representation offers a way to assess how species enter and leave the community, and how they change their interactions over time. It therefore provides a valuable way to empirically interrogate three stages of community dynamics: assembly, intermediate dynamics, and disassembly[22].

There are several hypotheses on how each of these three stages might progress in natural communities. Within-season assembly of plant-pollinator networks seems to be described by 'preferential attachment', the mechanism by which newcomer species are more likely to attach to generalist ones[1,23]. However, this preferential attachment hypothesis is not without contention, as contrasting mechanisms seem to also explain the assembly process in other mutualistic systems[24] and time scales (e.g. 'opportunistic attachment'[19]). The way communities disassemble, on the other hand, has been less frequently studied. Although some studies have shown the disassembly process in plant-pollinator communities to showcase the preferential loss of less-connected species (i.e. 'preferential detachment'[5,20]), how disassembly plays a role within seasons is often unexplored. Finally, the bridge between community assembly and disassembly—its intermediate dynamics—is largely missing (but see Tylianakis et al.[25]). The non-random structure of mutualistic communities suggest, nevertheless, a coherent dynamic in the way species enter the community, change their interactions, and leave the community.

Perhaps one of the main obstacles for untangling this coherent dynamic is finding the 'appropriate' scale. For example, some studies have focused on the change of species composition over time, adopting a 'full-network' perspective to community dynamics. Unfortunately, while it appears useful to explain observed species distributions[26,27], these studies often need to assume that interactions are independent from local changes in species abundances, thereby washing away a key component of community dynamics. Alternative approaches have found success quantifying temporal interaction turnover and linking such turnover to species' phenologies[14,28]. Nevertheless, these 'species-level' approaches are often centred around quantifying variation

of species interactions and lack the resolution to understand how such variation transforms the overall structure of ecological networks. That is, changes in species composition or interactions might not always translate into meaningful changes in the community structure (or vice versa).

We employ an approach here to study the complete seasonal dynamics of plant-pollinator communities using the technique of network alignment[29]. Conceptually, aligning any two ecological networks proceeds by pairing up the species that play similar structural roles in each community[21,30,31]. This pairing essentially identifies species with analogous 'positions' across communities (Fig. 1)—i.e. species that are similarly embedded in the corresponding network of interactions. It also offers a suitable scale to study community dynamics, one in which the state of any given species is always defined relative to all the other species in the community. This scale allows us to synthesize the information encoded within network time series, providing a comprehensive conceptual mapping of the changes in the communities and their many components.

In our study, we first use network alignment to assess the extent to which the positions of individual plant and pollinator species are variable within seasons. That is, given the alignment between the network observed in a community at two points in time, we use the information about who gets paired with whom to reveal whether and how species change their positions over time. We then evaluate the similarity of these positions across all of the data. In particular, we identify the set of distinct groups of species' positions found across networks, representing the characteristic ways in which species tend to be embedded in their community. This allows us to synthesize the complex dynamics of individual species over time into something much simpler: the movement of species across groups of positions within the network. Characterizing this movement, we display a road map on how plants and pollinators first enter, then comprise, and ultimately leave the networks. This enables us to prune down the seasonal assembly and disassembly processes in empirical plant-pollinator communities, respectively. Likewise, this road map allows us to reveal the mechanisms by which species vary their positions in the community over the course of a season, which species will stay in the network the longest, and what positions species occupy before leaving the community. Overall, our study uncovers the underlying structural dynamics of plant-pollinator communities, leading us towards general rules regarding species' interaction turnover, phenology, and assembly/disassembly processes in empirical plant-pollinator communities.

## Results

**Changes in species' positions over time.** We studied the interaction changes happening to an empirical plant-pollinator community over the course of three sampling seasons—i.e. examining three network time series composed of weekly interaction networks (Methods). We first focused on the study of species' positions to understand how these change over time. To do so, we aligned every pair of weekly networks in our dataset multiple times to identify the optimal alignments between them, uncovering the pairs of species sharing analogous positions (Fig. 1). Specifically, we analysed (i) the uniqueness in species' positions within networks and (ii) the variability of these positions across networks. An analysis of the uniqueness of species' positions can reveal how many species share the same position within any individual network and is an important measure as it can unveil the internal symmetries in the structure of these communities (Methods). An analysis of the position variability across networks can instead reveal how much the position of any given species

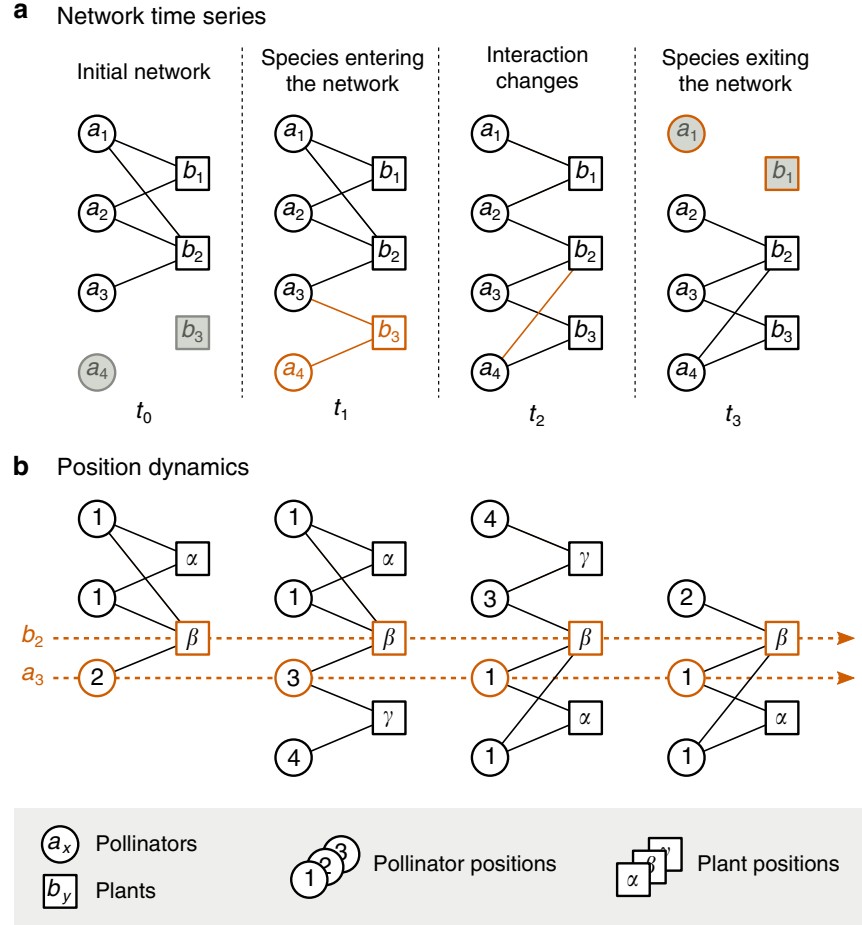

**Fig. 1 Dynamics of a bipartite network. a** An example of a time series for a plant-pollinator network. The circles and squares represent pollinators and plants, respectively. The links characterize interactions between these species. The colored species and links identify the changes made to the network over time. **b** The change in species' positions in the network time series represented in **a**. The different numbers describe different pollinator positions, and the different Greek letters describe different plant positions. The colored dotted lines indicate the position of two specific species $a_3$ and $b_2$. On the one side, species $a_3$ change its position over time, starting in position '2' and ending up in position '1'. On the other, species $b_2$ preserves the same position '$\beta$' over time. Note how several species can have the same position in the network.

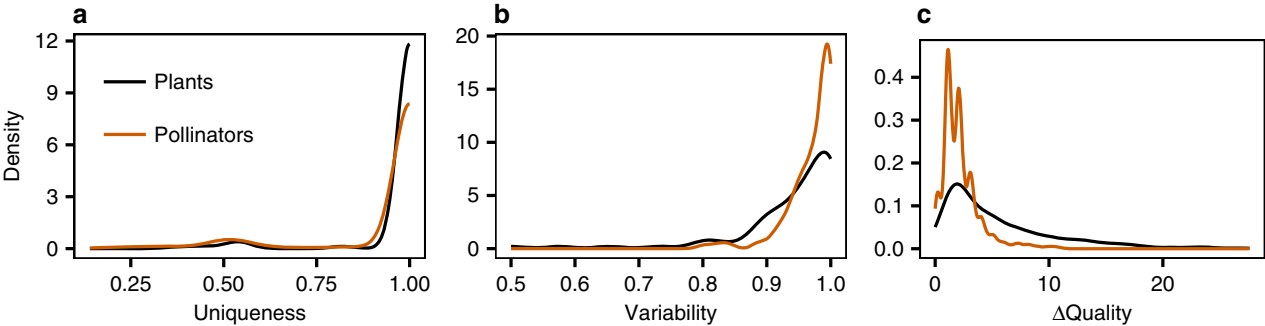

**Fig. 2 Analysis of species' positions across networks. a** Uniqueness of the position of plant and pollinator species within networks. **b** Variability of species' position across networks, including only plant and pollinator species that appear in multiple networks during a season. **c** Decrease in alignment quality due to the fixing of individual species that are common across networks.

changes across networks, shedding light on the structural dynamics of species over time (Methods).

The study of the uniqueness of species positions within networks showed that, in general, species tend to be paired to themselves, indicating the presence of very few symmetric positions within any given network (Fig. 2a). That is, species are uniquely positioned within networks (i.e. at each point in time). This is true for both plants and pollinators—with 93% of plant and 87% of pollinator positions presenting uniqueness values higher than 0.9 (Supplementary Note 1). Importantly, the same alignments performed using binary networks showed the uniqueness of species positions to be much lower in this type of networks (Supplementary Fig. 1). That is, we found that interaction strengths add a crucial layer of information to distinguish between species' positions in ecological networks.

Given that the positions of species within any network tend to be unique, we next studied the variability of each species' position over time. We found that nearly all species that appear in multiple networks tend to change positions, presenting very high position variability from network to network (Fig. 2b). That is, species' positions drastically change within seasons. Again, this was true for both plants and pollinators—with 81% of plant and 96% of pollinator positions presenting variability values higher than 0.9. We also found this to be true when considering binary networks (Supplementary Fig. 1). We then altered the alignment algorithm to artificially fix the pairing of common species over time, studying the effect of any individual species preserving its position over time (Methods). Somewhat more surprisingly, we found the fixed plant species to align worse than the fixed pollinator species. This suggests that plant species change their positions more drastically than pollinators over time (Fig. 2c), which could reflect plants' longer phenologies[32].

**Groups of species with similar positions**. The observed variability of the species' position over time hints at the degree of complexity that the dynamics of plant-pollinator networks encompass. As an attempt to reduce this complexity, we studied the similarity of all species' positions in our dataset. To do so, we compiled all alignments between networks into an alignment matrix $M$ describing who is paired with whom within and across networks (Methods; Supplementary Methods).

Using this alignment matrix, we tested whether or not there are fundamental groups of similar positions across networks (Methods). We found evidence to support the idea that there are three distinct groups of pollinator positions and three distinct groups of plant positions (Fig. 3a). Then, we characterized the nature of these distinct groups by measuring basic node properties for each of them (Fig. 3b; Methods). In particular, we used measures of species' relative degree—the number of different interactions of a given species divided by the number of interactions of the most connected species of a network—because these measures are suitable for comparing species from differently sized networks. Focusing on the three groups of pollinator species (A, B and C in Fig. 3), we found that these show well-defined differences: group A represents species with low degree that interact with at least one generalist plant species; group B represents species with high degree that also interact with at least one generalist plant species; and, group C represents species with low degree and that interact with low degree plant species (see effect sizes for the comparison of position groups in Supplementary Fig. 2; Methods; Supplementary Methods). These results were consistent across seasons (Supplementary Figs. 3 and 4), and we found similar results for the plant species in the networks (Supplementary Figs. 5–7). In addition, the groupings showed a strong agreement with the results found using different valid community detection methods (see Supplementary Table 1 and Supplementary Fig. 8). Although we lacked completely independent measures of pollinator abundances to properly test their influence on the observed patterns, we studied the relationship between species' relative degree and their abundances using flower counts for plants and estimates from interaction data for pollinators (Supplementary Note 2). While we found a positive relationship between these factors, a linear regression showed considerable variation in species' relative degree that is not explained by abundance (85.9% and 66.5% for plants and pollinators, respectively; Supplementary Note 2; Supplementary Fig. 9). This implied that abundance alone cannot fully explain species' memberships in the three groups.

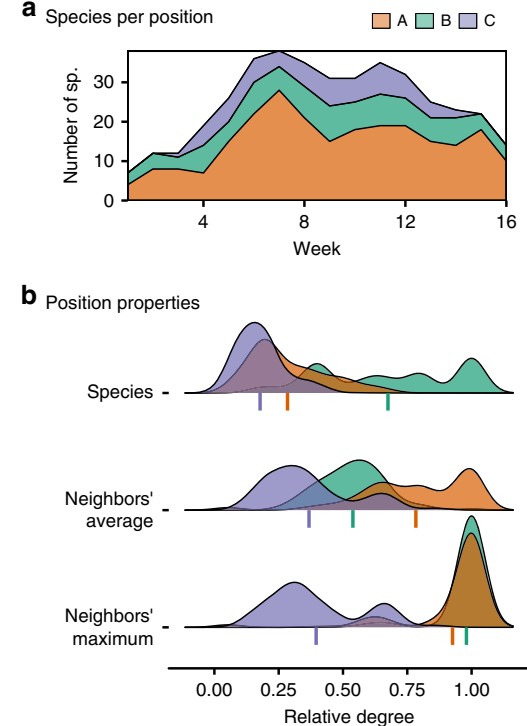

**Fig. 3 Distinguishing properties of the distinct groups of pollinator positions. a** Number of pollinator species in each group of positions over time. Each color represents a different group of positions. **b** Different properties summarizing the species forming each group. The top panel shows the relative degree distribution of species in each group. The middle panel shows the neighbors' average relative degree for the species in each group. The bottom panel depicts the relative degree distribution of the most connected neighbor of every species in each group. The colored segments depicted under the distributions characterize the mean of each distribution. The results displayed correspond to the 2015 sampling season, as this is the year for which we had the highest number of records. Similar results for the 2013 and 2014 sampling seasons can be found in Supplementary Figs. 3 and 4.

**Movement of species across position groups**. Finally, we modelled the movement of the species across the different groups of positions found in the alignment matrix using a multinomial logistic regression (Methods). In particular, we estimated the time-dependent transition probabilities for the species moving across groups. These probabilities characterize how likely it is for any given species to change positions in the network, moving from one position group to another—e.g. the likelihood of a specialist pollinator that interacts with generalist plants (group A) changing interactions to become a generalist pollinator interacting with generalist plants (group B). We found that the results showcase a characteristic dynamic underlying plant-pollinator networks (Fig. 4). Moreover, the lower performance of models that treat the different seasons as independent temporal replicates suggests that this dynamic is also consistent across seasons (Supplementary Methods; Supplementary Fig. 10). Given the three groups of pollinator species found across networks, this dynamic can be described as follows: the positions of species entering the network tend to be from group A (specialist pollinators interacting with generalist plants); once in group a, these species tend to either stay in the same group, exit the network or move to group B (generalist pollinators interacting with generalist plants); species entering group B tend to either stay in that group or move back to group A; and species entering group C (specialist

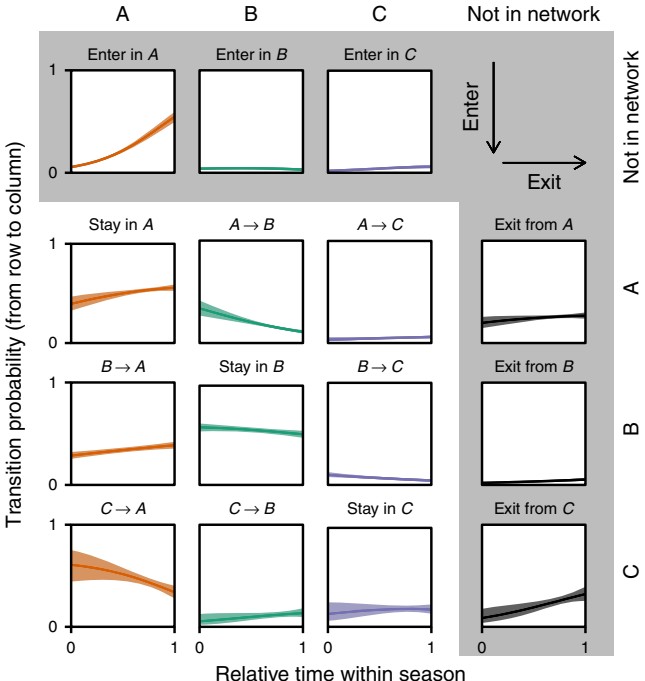

**Fig. 4 Movement of pollinator species across position groups.** Every panel in the matrix describes the inferred transition probability between the different groups of positions as a function of the relative time within the season. The order of the matrix is such that it characterizes the transition probabilities from row groups to column groups over time. The different groups are those presented in Fig. 3. The shaded row describes the probabilities of species entering the network into the different groups (A, B and C) whereas the shaded column describes the probabilities of species exiting the network from each of these groups. The different lines in the graph represent the mean transition probabilities and the shade of each line characterizes the first and third quantiles. Notice that the color is chosen based on the recipient group of the transition probability.

pollinators interacting with specialist plants) either move to group A, exit the network or stay in group C (see the effect sizes for the comparison of these transition probabilities in Supplementary Fig. 11; Supplementary Methods). Notice that similar results were found for the plant species in the network (Supplementary Fig. 12). In addition, similar results were also found using alternative probabilistic models that considered constant transition probabilities (Supplementary Methods; Supplementary Fig. 13).

## Discussion

In this work, we use a network-alignment technique as a way to disentangle the seasonal dynamics of plant-pollinator networks. First, we studied the uniqueness and variability of species' positions within and across networks, respectively. We found that species have unique network positions at every time point, but they also tend to change such positions over time. Assessing the similarity of positions over time, we then found that there are major groups of positions characterizing plant-pollinator communities. These groups of positions provide a suitable scale to synthesizing complex ecological dynamics. For pollinator species, for example, they can be broadly described as: (i) specialist pollinators that interact with at least one generalist plant, (ii) generalist pollinators that interact with at least one other generalist plant, and (iii) specialist pollinators that interact with other specialist plants. Using these groups of positions, we estimated the underlying dynamics of species within seasons and found

general rules regarding species' seasonal dynamics within plant-pollinator communities. Putting this all together, our results suggest that the structure of plant-pollinator networks is extremely dynamic, where species rapidly switch positions within the network over a season. This structural dynamic, however, is also coherent across years, and one can predict the changes in species' positions within networks over time.

The study of network time series is challenging due to the many levels of information that these systems encompass. One could, for example, adopt a full-network perspective and study community dynamics using general network metrics[33]. Unfortunately, network metrics lack the resolution to distill the mechanisms by which species change positions over time[19]. Indeed, the study of metrics such as nestedness and connectance has shown certain mutualistic networks to exhibit generally constant structures over time[15,34,35] (but see CaraDonna and Waser[32]). While this may be useful for understanding their dynamical stability and functioning[10,36,37], these metrics are particularly ill-suited to understand the full scope of plant-pollinator seasonal dynamics. Alternatively, one could use single-species approaches. Ecological data, however, are often clouded by environmental variability[38,39] or sampling errors in the data collection[40], both of which can add considerable noise to single-species dynamics. Perhaps most importantly, these approaches could also easily be overwhelmed by species' natural idiosyncrasies[41], which could mask potential general rules governing community dynamics. Indeed, we observed the effects of such idiosyncrasies when studying the uniqueness and variability of species' positions. The high uniqueness of species' positions indicates how singularly different species are embedded within a network; and, the high variability shows how sensitive these positions are to changes in the network structure. Noticeably, our observations on the variability of species positions also agree with recent work showcasing constant temporal switching of species' interactions in empirical plant-pollinator communities[14,42]. Therefore, the commonly used static network representation, albeit useful in many cases, might strongly constrain our understanding of the dynamic nature of some ecological communities. Much like the artificial nature of the geographic boundaries between networks[43], one could argue that temporal boundaries are just as artificial[44].

Here, we illustrate how it is possible to find a useful middle ground between full-network and single-species approaches. In particular, we focused on identifying distinct groups of positions within networks by clustering species with similar positions. This group scale allowed us to strategically prune down plant-pollinator dynamics. Assessing the movement of species across these groups of positions, one could, for example, focus on how pollinator species enter the community. As expected, we found the degree of newcomers to be generally low; we observe species entering the community mostly as specialists (group A from Fig. 3). These new-coming species tend to interact with at least one generalist plant, showing consistency with the idea of preferential attachment[1]. Likewise, the detachment of pollinators from networks often comes from groups of less-connected species (groups A and C from Fig. 3), also in agreement with the idea of preferential detachment[5,20]. The symmetry between these two processes—preferential attachment and detachment—has been showcased at longer time scales[25] as well as hypothesised to generate and maintain network patterns promoting stability[22]. For example, Tylianakis et al.[25] used simulations to show how commonly observed nested patterns can arise as a result of such symmetric processes.

Our results provide a road map for how species change positions within the community, which positions are those that species take before exiting the network, and which species will

likely remain in the network the longest. For example, our approach reveals that once in group A (specialists interacting with generalists), a species will often remain in this group, or simply leave the network. This pattern is observed with *Anthophora terminalis* (orange-tipped wood digger bee), which briefly appears in the network across years as a specialist pollinator interacting with generalist plants. Other species, however, may change their interactions in such a way that moves them from one group to the next (e.g. from group A, specialists interacting with generalists, to group B, generalists interacting with generalists). Species in group B will either stay in this group, or move to group A (rarely do we observe species in group B leaving the network). *Bombus bifarius* (two-form bumble bee) is such an example, and is often found in the networks as a generalist pollinator interacting with generalist plants. Finally, our results also reveal that when species are in group C (specialists interacting with other specialists), they will most likely move to group A or leave the network; *Arctophila flagrans* (a flower fly) showcases these transitions across years.

Overall, we found species' network-position dynamics to be consistent across the three different sampling seasons from our dataset. In other words, these general patterns we have reported—i.e. the way in which species first enter, then comprise, and ultimately leave the networks—are consistent from one year to the next. This is important because it showcases how network-position groups could be used as fundamental building blocks for understanding plant-pollinator community dynamics (similar to the concept of trophic components in predator-prey food webs[45,46]). For the sake of simplicity, we chose the 'short random walks' algorithm to identify these position groups, as it provided us with a conveniently sized grouping of species positions. However, different community detection methods can partition the alignment matrix differently, providing different degrees of resolution to the dynamics of species across groups. Although, we showed how more complex partitions display more resolved dynamics (Supplementary Fig. 8), we also found that finer resolutions might lead to groups of positions that can be difficult to discern from each other in purely ecological terms (Supplementary Fig. 14).

There are multiple factors that could play a role in explaining plant-pollinator dynamics beyond species' degree. Information on species' abundances is likely one such factor, as species' phenologies have already been linked to interaction turnover in this system[14], and abundances are good predictors of network structural properties in other ecological systems[47,48]. However, the evolutionary fingerprint underlying empirical communities[49] and evidence regarding pollinators' plant preferences[50,51] point towards the idea that species' abundances cannot represent the full picture regarding species' interaction dynamics[52]. In some pollinator systems, for example, the pollinator costs of searching for trait-matching resources has been shown to be lower than the cost of switching to more abundant ones[53,54]. Indeed, species' abundances are often considered emergent population-level properties that are ultimately constrained by species' traits, and these same traits have been shown to effectively predict empirical plant-pollinator interactions[55]. That being said, the link between species' abundances and networks' structural dynamics define two ends of an interesting conceptual spectrum: one in which interaction variability can be explained solely based on species' abundances, and the other where species interaction changes are completely independent from their abundances. Previous research indicates that empirical communities likely fall somewhere in between these two cases[6,25,56], and our results also seem to support this idea. While pollinator observations and flower counts here showed positive relationships with species' relative degree, these weak correlations left a lot of variation unexplained (Supplementary Fig. 9).

Finally, we identify three areas we feel represent key steps from which to move forward. First, the approach used in the present work is not limited to plant-pollinator networks. Indeed, it could be used to shed light on the mechanisms governing many other systems, including food webs[57], host-parasite communities[58] or other types of temporal networks[59]. Second, while we focused on temporal variation, another interesting perspective would be to put the same tools to work across other type of gradients[60]. For example, one could focus on the structural variability of plant-frugivore networks along forest-farmland gradients[3], which could reveal how bird species change positions within networks in order to adapt to different environmental conditions. Third, we defined species' positions purely based on the structure of plant-pollinator communities. Nevertheless, these positions could easily also account for other species' properties such as species' ecological traits and evolutionary histories[29]. This would allow us, for example, to study network dynamics from a functional diversity or evolutionary perspective, potentially untangling the eco-evolutionary mechanisms governing complex community dynamics.

## Methods

**Empirical data**. We studied plant-pollinator interaction networks from a subalpine community in the Colorado Rocky Mountains (USA)[14,61]. These data were sampled at weekly intervals over three summer growing seasons, and contain nearly 30,000 pairwise interactions between a total of 93 pollinator species and 46 flowering plants. To study the dynamics of these plant-pollinator communities, we aggregated the observed interactions into weekly plant-pollinator networks, where the weight of all interactions was set to the absolute number of observed interactions between the corresponding species pair during that week. In total, this resulted in three seasonal network time series comprising 12, 15, and 16 weekly weighted networks, respectively (see CaraDonna et al.[14] and CaraDonna and Waser[32] for further details). These plant-pollinator networks have been shown to be robust to sampling effort, with interaction rarefaction curves and abundance-based richness estimates indicating consistent sampling and an average detection of interaction of 85–93% across weeks (see Supplementary Fig. 1 and Supplementary Table 2 from CaraDonna et al.[14]). Note that although we focused on the study of weighted networks (also referred to as quantitative network), we also considered their binary counterparts for different aspects of this study. In a binary network (also referred to as qualitative network), the link between two species exclusively indicates the presence of an interaction between them, and the strength of the interaction is ignored. We therefore generated binary networks by setting the weight of all interactions in the weekly plant-pollinator networks to a common value of 1.

**Network alignment**. To analyse the dynamics of these network time series, we used the alignment technique introduced by Bramon Mora et al.[29]. This technique provides us with a way to map two ecological networks on top of each other. Given two networks A and B, network alignment pairs up the species $i \in A$ and $j \in B$ together using the 'structural roles' that they play in their respective communities. While there are multiple ways in which one could define these structural roles, we based our definition on the concept of network motifs[62]—i.e. the set of distinct patterns of interactions between $n$ species found within a network. Following the ideas presented by Stouffer et al.[30] and Baker et al.[31], we defined the role of any species based on the number of times it appeared in any of the distinct positions found within motifs made of 3, 4 and 5 species. In particular, we used the tools developed by Bramon Mora et al.[63], where species' structural roles account for information regarding the strength of their interactions.

As described in Bramon Mora et al.[29], aligning networks is a stochastic optimisation process, where multiple random alignments $\lambda$ between the species in A and B are proposed in order to find the optimal pairing between these species' roles—i.e. the optimal alignment $\lambda^*$ between A and B. As a result, this optimal alignment provides us with three key pieces of information: (i) the optimal species-species pairing between all species $i \in A$ and $j \in B$; (ii) a cost function $C_\lambda$ characterizing the similarity between A and B; and (iii) the role correlation $c_{ij}$ of every species-species pairing. On the one side, the species-species pairing identifies species that are similarly embedded within their respective networks, since the alignment pairs up species that occupy similar positions across networks. On the other, $C_\lambda$ and $c_{ij}$ describe the 'quality' of the alignment and each species-species pairing, respectively.

Notice that aligning networks can produce multiple equally valid solutions. For example, a given set of $n$ alignments $\{\lambda\}$ between A and B can reveal multiple species-species parings that minimize the cost function $C_\lambda$. Therefore, one should align any pair of networks multiple times in order to properly compare their structures and uncover as many pairs of species that share analogous positions

across networks as possible. Likewise, as noted by Bramon Mora et al.[29], the alignment algorithm provides a one-to-one species pairing; therefore, there will necessarily be species that remain unpaired when comparing networks of different sizes. While the pairing of two species indicates their analogous position, an unpaired species indicates its singular position (see Bramon Mora et al.[29] for further details and tests of the alignment algorithm).

**Position uniqueness within networks**. To analyse whether or not species' positions were unique, we separately compared the structure of the community at each time point. If a given species $i$ has a unique position within a network A, the alignment of A with itself should always pair up $i$ with itself (e.g. the position of species $b_2$ is unique at any point in time in Fig. 1b). In contrast, if $i$ has a position that is not unique, repeating the same alignment should result in different pairings for species $i$—also pairing $i$ with all species within A that share the same position (e.g. species $a_3$ in Fig. 1b shares position with $a_4$ at $t_2$ and $t_3$). Accordingly, we aligned every weekly plant-pollinator network in our dataset to itself 100 times. This allowed us to identify the distinct pairings of species that produced optimal alignments. Given the 100 independent alignments, we then measured the uniqueness of the position of a species in a network as the proportion of these alignments in which a species was paired to itself.

**Position variability across networks**. To measure the variability of species' positions over time, we compared networks at different time points. Specifically, we aligned every pair of networks in a given season 100 times and analysed species that are common to any of these pairs. For example, given two networks $A_{t_1}$ and $A_{t_2}$ that were collected at two time points $t_1$ and $t_2$, we wanted to test whether or not any species $i$ present in both networks—i.e. a species $i \in A_{t_1} \cap A_{t_2}$—changed its position over time (e.g. species $b_2$ and $a_3$ in Fig. 1b). To do so, we looked at whether or not species $i \in A_{t_1}$ was paired to species $i \in A_{t_2}$ in the alignment between $A_{t_1}$ and $A_{t_2}$ (e.g. species $b_2$ retains its position $\beta$ over time while species $a_3$ changes positions in Fig. 1). Following this, we measured the position variability of a species $i$ as the probability of $i$ being paired to a different species $j$ in any alignment between networks containing $i$.

**Measuring the change of position of individual species**. We then measured how much the positions of individual species changed over time. To do so, we re-aligned the networks while artificially fixing the pairing of common species. That is, for any pair of networks $A_{t_1}$ and $A_{t_2}$ with $n$ common species $i$, we performed $n$ alignments in which we individually fixed the pairing of each species $i$ and freely aligned the rest. The difference in the quality of the alignment due to the fixing of any $i$-$i$ species pairing can be used to measure $i$'s change of position. For example, if the quality of the alignment changes drastically when fixing the pairing $i$-$i$, it means that species $i$ has significantly changed positions from $t_1$ to $t_2$. In contrast, fixing the pairing $i$-$i$ will not change the alignment quality if species $i$ presents the exact same position in $A_{t_1}$ and $A_{t_2}$. Following this, we used this comparison across all alignments between the networks in our dataset to reveal the effects of fixing individual species' pairings, focusing on the differences between fixing plant and pollinator species.

**Alignment matrix**. Two species $i \in A$ and $j \in B$ could have very similar positions and still not be paired in the alignment between A and B. This would happen, for example, if there was a third species $k \in B$ that also had an identical position to $i$—in which case species $i \in A$ would be paired to species $k \in B$. The similarity between $i$ and $j$, however, can be studied by aligning the networks A and B to other networks $\{C, D, \ldots\}$. In doing so, we would likely observe some degree of overlap between the pairings of $i$ and $j$ across these other networks. This overlap would indicate that $i$ and $j$ share similar positions.

To uncover these types of similarities between all species' positions in our dataset, we compiled here all alignments into an alignment matrix $M$ (Supplementary Methods). In this matrix, every element $m_{ij}^{AB}$ accounts for the pairings between any two species $i$ and $j$ from any two given networks A and B, respectively. Specifically, it accounts for how often these two species are paired following a set of 100 alignments $\{\lambda\}$ between A and B as well as for the quality of such pairing. Notice that $M$ contains information regarding all alignments between all networks in our dataset. Therefore, every row or column of this matrix represents a species $k$ of a given network $X$, describing all its pairings within and across networks.

**Identifying distinct groups of species' positions**. The alignment matrix $M$ allowed us to analyse the similarity across all species' positions in our dataset. In particular, we focused on identifying distinct groups of species' positions that were similar within and across networks. To do so, we used a 'short random walks' algorithm to identify the 'modules' within the matrix $M$ describing sets of species' positions that align more often with each other than they do with the rest[64]. Notice that other community detection methods can produce other valid groupings (see Supplementary Table 1). Therefore, we used a normalized mutual information analysis to study the agreement of the 'short random walks' algorithm with other

community detection methods designed to analyse large weighted undirected graphs such as the alignment matrix used here[65] (Supplementary Table 1).

Given the grouping of species with similar positions in our dataset, we described the different groups using basic information about the way their constituent species interact with each other. We focused on measures of species' relative degree $k_i = l_i/l_{\max}$, where $l_i$ is the number of qualitative interactions of a given species $i$, and $l_{\max}$ is the number of qualitative interactions of the most connected species of the network. A sensitivity analysis of species' relative degree for plants and pollinators showed this measure to be robust to sampling effects, with relatively small absolute changes in the metric following the random removal of observations (Supplementary Fig. 15). We also observed a moderate average loss of qualitative interactions following this removal (below 30% loss for 50% removal; Supplementary Fig. 15). Consider a network formed by a set of species with relative degree $\{k\}$, and let $\{k^i\}$ be the set describing the relative degree of the $k_i$ interacting partners of species $i$. For any group of species $G$, we studied: (i) the relative degree of the species in the group, $k_i \forall i \in G$; (ii) the relative degree of the most connected interacting partner of each species in the group, $\max\{k^i\} \forall i \in G$; and (iii) the average relative degree across all interacting partners of every species in the group, $\langle\{k^i\}\rangle \forall i \in G$.

To statistically compare the properties of species from different groups, we used a multivariate autoregressive model. In particular, we considered the three node properties described above (i, ii and iii) as response variables, and estimated them using species' group, accounting for sampling season, species identity, temporal autocorrelation within each year, and residual correlation among position properties (Supplementary Methods). We then used the resultant model to study the effect sizes for the relationships between group and position properties (Supplementary Methods).

**Species structural dynamics across groups**. We examined the movement of pollinator and plant species across the different network positions using a probabilistic model. Given $n$ groups of positions, our model describes a scenario in which every species can be found in $n + 2$ possible states $\vec{y}$ at time $t$. These include: $n$ states $y_1 \ldots y_n$ characterizing a species in each of the different groups of positions; a state $y_{n+1} = y_{\mathrm{pre}}$ describing a species that has not yet entered the network; and a state $y_{n+2} = y_{\mathrm{post}}$ describing a species that has already exited the network.

We used a Bayesian multinomial logistic regression as a way to estimate the rate of movement between the different states of the species over time[66,67]. In particular, our model considers $n + 2$ types of events, describing the transition probability of species from any state $y_i$ to any possible state $y_j$ at time $t$ as:

$$\Pr\left(y_j | s_1(y_i, t), s_2(y_i, t), \ldots, s_{n+2}(y_i, t)\right) = \frac{\exp s_j(y_i, t)}{\sum_{k=1}^{n+2} \exp s_k(y_i, t)}, \quad (1)$$

where $s_k(y_i, t)$ are 'scores' that determine the resulting probabilities. To infer these scores, we first fixed one of them to serve as an arbitrary baseline—e.g. assigning $s_1(y_i, t) = 0$. We then estimated the remaining scores as $n + 1$ linear models of the form

$$s_k(y_i, t) = \alpha_{k1} + \beta_{k1}t + \sum_{l=2}^{n+2}(\alpha_{kl} + \beta_{kl}t)\delta_{il}, \quad (2)$$

where $\alpha_{kl}$ and $\beta_{kl}$ are the parameters inferred by the model, and $\delta_{il}$ is a Kronecker delta that is set to 1 if $y_l = y_i$, and 0 otherwise. The time variable $t \in [0, 1]$ is defined relative to every sampling season—i.e. $t$ is calculated as the week number divided by the total number of weeks in each sampling season. We used the R package 'rstan' to generate the posterior samples for the Bayesian models[66,68]. For all parameters $\alpha_{kl}$ and $\beta_{kb}$ we chose weakly informative normally distributed priors, with mean $\mu = 0$ and standard deviation $\sigma = 10$ (Supplementary Methods). Importantly, species are assumed not to re-enter the network once they have exited it, making the transition from state $y_{\mathrm{post}}$ to any other state occur with probability zero. We also treated species phenologies as uninterrupted; therefore, we considered any observation of a species transition from any state $y_i$ (i.e. species in the network) to $y_{\mathrm{pre}}$ (i.e. species not yet in the network) and back to any state $y_j$ during its activity period to be the result of a likely sampling error. Note that we ignored any of such observations (i.e. $y_i$ to $y_{\mathrm{pre}}$ and $y_{\mathrm{pre}}$ back to $y_j$) when inferring the probabilities.

While we estimated the time-dependent transition probabilities analysing all sampling seasons together, we also considered multiple other forms for the multinomial logistic regression. Specifically, we considered models accounting for the different sampling seasons as temporal replicates in different ways as well as models with constant transition probabilities over time (Supplementary Methods). Finally, we used the widely applicable information criteria (WAIC) to inform the relative support of the different models. This comparison allowed us to understand the effect of accounting for the different sampling seasons as replicates, testing whether or not the estimated probabilities are consistent across years.

**Reporting summary**. Further information on research design is available in the Nature Research Reporting Summary linked to this article.

## Data availability

The primary data associated with this manuscript are available in the Environmental Data Initiative (EDI) digital repository[61].

## Code availability

Code to conduct the network alignment can be made available upon request.

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

## Acknowledgements

For their help and discussions on the project, we thank the members of the Stouffer and Tylianakis Labs. Special thanks go to Fernando Cagua for crucial discussions on the manuscript and inspiration for the figures. B.B.M and D.B.S. acknowledge the support of a Rutherford Discovery Fellowship (to D.B.S.), administered by the Royal Society of New Zealand. E.S acknowledges the support of the Singletary scholarship, the UKY Education Abroad Scholarship and the honours college education abroad grant, all administered by the University of Kentucky. P.J.C. acknowledges the support of the NSF grant DGE 11-43953 (to P.J.C.).

## Author contributions

B.B.M. led to the design of the work; contributed to the writing of the code to perform the network alignment; performed the research; led the writing; and gave final approval for publication. E.S. contributed to the writing of the code to perform the network alignment; contributed to the revisions; and gave final approval for publication. P.J.C. provided the data; contributed to the revisions; and gave final approval for publication. D.B.S. contributed to the design of the work; contributed to the writing of the code to perform the network alignment; contributed to the revisions; and gave final approval for publication.

## Competing interests

The authors declare no competing interests.
