## [Peer Review File · Nature Communications]

Reviewers' Comments:

Reviewer #1:

Remarks to the Author:

Bramon Mora et al. use a network alignment method to identify rules governing the three stages of community dynamics (assembly, intermediate, and disassembly) of a highly resolved plant-pollinator network in the Rocky Mountains of Colorado. I think the authors' unique methods and their simultaneous assessment of different stages of community dynamics have merit and would contribute to a more general understanding of the temporal dynamics of ecological networks. Still, I have many comments that I hope the authors will find constructive.

****Interpretation****: The authors suggest that species entering and exiting the community tend to be specialists, an inference based on the low relative degree of groups A and C (Fig. 3). However, do these species have a low relative degree because they are simply less abundant? If that's true, then the assembly and disassembly process is simply due to species increasing and decreasing, respectively, in abundance over the course of the season. This would not be surprising and perhaps be a trivial result. It would help if the authors provide a clear definition of how degree and relative degree were quantified with their weighted network data as well as the sensitivity of these network properties to sampling effects (i.e. abundance). In terms of the intermediate dynamics, it appears that there are more patterns to unpack (e.g. common transitions include: stay in A; B->A or stay in B; C->A); however, the biological significance of these "intermediate dynamics" were not discussed much, despite this being framed as one of the novel contributions of this study (Intro. L. 56-58).

****Statistical evidence****: Despite the several assertions made in regard to identifying general assembly rules, there is no statistical support for any of these conclusions. For example, the authors describe "the position of species entering the network tend to be from group A", but there is no evidence that this is "above average" or different than another group. They should be able to infer this information from their Bayesian model (which also needs further description, see comments below) and should report this in the Results. The authors should also provide statistical evidence supporting their descriptions of groups A, B, and C (I've provided one possible way to do this below).

****Bayesian models****: The Bayesian models are not described well at all. What priors did the authors use? What statistical software did they use (e.g. they cite McElreath 2018, but it's unclear whether they use his "rethinking" package in R). Why don't the authors report any statistical evidence to support their assertions about the assembly rules they claim to have discovered (e.g. effect sizes with credible intervals for each group's transition probability). They could also consider using widely applicable information criteria (WAIC) or leave-one-out cross-validation information criteria (LOOIC) to inform the relative support of the different models they analyzed (e.g. transition probabilities are time-dependent, constant, or density-dependent). The authors could then use these criteria to choose which model to highlight in the main text, rather than make what appears to be an arbitrary choice. For example, eye-balling the quantiles in Fig. S9 vs. Fig. 4, makes me think that the constant probability model may be the best choice (smaller quantile range and fewer parameters; in fact, it seems like the authors' interpretation of assembly rules follows more from the constant probability model rather than the time-dependent one which is highlighted in the main text..). On a related note, why was each year analyzed separately? Why not treat each year as a temporal replicate and test whether the transition probabilities differ between years (which would suggest assembly rules depend on the year, i.e. there are no general rules) or are consistent across years (i.e. there may be general rules for this community).

****Identifying node-level properties of each group****: If you're comfortable using Bayesian models, one possibility would be to use a multivariate autoregressive model with the brms package in R.

```
Sample code: brm(mvbind(species, neighbours-average, neighbours-maximum) ~  
year + group + (1|species),  
autocor = cor_ar(formula = ~ week, p = 1)).
```

This could provide you with information on the effect sizes and credible intervals for the relationship between group and node properties that also accounts for confounding factors (e.g. year, species ID, temporal autocorrelation within each year, and residual correlation among position properties). This approach would also allow you to use all of the information, rather than separating it among years. Note that this is a rough sketch and you may find a better alternative. The key point is that you should show statistical evidence for the claims you make.

****Static networks****: I feel the authors build a bit of a "straw man" argument in the Intro and Discussion regarding theory of "static interaction networks". These networks are only static in regard to per-capita parameters (e.g. intrinsic growth rates, inter- and intraspecific interaction rates, handling time, to use Bastolla et al. 2009 as an example). In contrast, population-level interaction strengths do dynamically change in these models as the abundance of plants and pollinators changes. If the authors showed, for example, that per-capita interaction strengths were dynamically changing over time, then I think there would be more room for critique. As is though, the dynamic nature of the plant-pollinator network analyzed here could simply be due to changes in abundances of interacting populations with constant per-capita parameters, which would be consistent with the assumptions of the "static interaction networks". Perhaps the "density-dependent" transition probabilities would give more insight here; however, the authors haven't provided any details on exactly how density dependence was included in that model.

Minor Comments

Overall, I appreciate the time and effort the authors put into producing an aesthetically pleasing and well-written manuscript.

Intro. L. 37: I would only use "evolve" here if you mean "biological evolution" (I assume you aren't referring to biological evolution though based on the references given). I would simply use "change" to be more clear.

Intro. Fig. 1: I really appreciate the clear description of the 'network alignment' methodology provided in this figure.

Intro. L. 95: "then comprise"? This word choice seems odd to me.

Methods (general): For the most part, I think the authors do a good job of clearly describing the "network alignment" methodology, which will be unfamiliar to most ecologists. For example, I thought the **Measuring the change of position of individual species** section was quite clear and had a good example (L. 178-183). Still, there were other places where I needed more help. For example, I got confused in the **Position variability across networks** section. It may be helpful to reference Fig. 1 (which is excellent by the way) at times to help more visual learners understand exactly what your methods are doing. For example, the authors could edit the sentence, "For example, given two networks At_1 and At_2 that were collected at two time points t_1 and t_2 , we wanted to test whether or not any species i present in both networks---i.e. a species i \in At_1 \cap At_2 ---changed its position over time (e.g. species a_3 in Fig. 1b)."

Methods L. 232-234: "Importantly, species are assumed not to re-enter the network once they have exited it, making the transition from state y_{post} to any other state occur with probability zero." Although this is a fantastic dataset, my guess is that it is not uncommon for a species to appear to

exit the network (e.g. not observed one week), and then "re-enter", even though the apparent exit from the network could just be due to sampling error. How was this situation treated? Were these treated as actual "exits" even though they probably were not?

Methods L. 206-216: Why did the authors focus on the 'short random walks' algorithm? Is there is a biological/statistical justification? Or is it because this algorithm produced three groups of plants and pollinators, which just felt like a good number of groups for interpretation...

Methods L. 208-211: Please provide a clear description of how you quantified degree and relative degree for this weighted network data.

Results Fig. 2b and Fig. 2c: Intuitively, I feel like the results for variability (Fig. 2b) and quality (Fig. 2c) should be the same, but apparently they give different insights (e.g. plants appear to exhibit less variability in their positions across networks, but the low quality suggests they exhibit more variability than pollinators). It would be worth clarifying here or in the Methods how these different ways to assess change in species position over time have different biological meanings.

Results L. 267: Why highlight the year 2015? It makes me think there is something special about this year, but it's just one of the temporal replicates right?

Results L. 281-286: This appears to be a description of the general assembly rules the authors discovered. I think it would be much more helpful for the reader if the authors describe this in biological terms (e.g. species entering the network tend to be specialists that interact with generalists that were already present (i.e. group A)).

Discussion L. 328-329: "Noticeably, our observations on the variability of species positions also agree with recent work showcasing constant temporal switching of species' interactions in other empirical plant-pollinator communities (CaraDonna et al., 2017; Miele et al., 2019)". One of these studies actually contributed the data to this study, so "other" is a bit misleading.

Reviewer #2:

Remarks to the Author:

This is a complex, but extremely well-written paper tackling a very relevant question in community ecology. Despite I am not familiar with the methods used, and those are complex, the paper is easy to follow. Most of my comments try to link the methods with the empirical observations to enhance the story.

My main concern, as always in network ecology, is about sampling effort, and the possibility of having unobserved links in the dataset. A couple of sentences about sampling coverage and how robust is your analysis to unobserved links may reinforce the conclusions.

A second recommendation would be to give a few more details on how the pairing works. Especially related to (i) how the structural roles are defined, which now is only in the supplementary material and (ii) how the pairing works when you compare matrices of very different sampling sizes. This is something that I still don't fully understand. For example, in the section starting at line 142, "species positions". It is unclear to me if the definition of uniqueness depends on the number of species within a network analyzed? I understand uniqueness is a fraction, which may be biased by the number of potential pairs that can be analyzed, but I am not sure. If unequal richness is an issue, how do you take into account species richness?

The following comments are most of them minor and In Line:

Line 116: "where the weight of all interactions was instead set to 1." I understand what you mean, but you can explain how you binarize the networks in a clearer way.

Line 232-234: This means NO species was sampled e.g. in week 2, not in week 3, and again in week 4? This is surprising.

Line 245: Given the comparison performed, I expected species to resemble the most to themselves (be paired with). Another potential question may be how close is the next potential pair is, right? In any case, I understand uniqueness is expressed in % and I understand this as the effect sizes of your analysis and I would like to see those written here.

Line 254: I agree that it changes across seasons, but given that weekly networks are most likely incomplete, I wonder how sensitive is your analysis to those sampling biases. For example, if you compare one network, with the same network jittered, how consistent are the positions? I would love to see that small perturbations to the interaction frequencies do not alter drastically the results.

Line 260. Are plant phenologies larger on average than pollinator phenologies? This may explain the discrepancy.

Line 283 I would suggest refreshing the reader that group A is a specialist- generalist interaction, and so on.

Line 300: I would suggest relating those positions to the nested structure of most networks, or to the concept of core species (generalist-generalist), peripheral species (specialist-generalist) and specialized interactions. Of course, a discussion on how abundance may be responsible for these observed patterns is warranted.

One key element which may help a lot more empirical readers to add more natural history to the Discussion. For example, show me an example of a real species and how it changes roles along the season.

Line 348: But how much this is driven by abundance patterns. The first and last individuals of any species are likely under-sampled for its interactions, and hence appear only in a small fraction of their potential interaction niche. Hence, what you are observing may be a side effect of a gaussian phenology-activity distribution and not an emergent property of the networks. I am just playing devil's advocate here but would be a discussion worth having.

Figure 4. Effect sizes look minor in several panels. This may be better reflected in text and mention the probabilities of each transition (and associated errors) explicitly.

Best,
Ignasi Bartomeus

Reply to Reviewer #1

Bramon Mora et al. use a network alignment method to identify rules governing the three stages of community dynamics (assembly, intermediate, and disassembly) of a highly resolved plant-pollinator network in the Rocky Mountains of Colorado. I think the authors' unique methods and their simultaneous assessment of different stages of community dynamics have merit and would contribute to a more general understanding of the temporal dynamics of ecological networks. Still, I have many comments that I hope the authors will find constructive.

Reviewer #1 stated the methods and ideas had merit and would contribute to a general understanding of plant-pollinator community dynamics. That said, the Reviewer also provided several comments on aspects in which the manuscript and analysis need to be improved in order to clarify the potential gaps as well as gain robustness. In particular, the Reviewer offered valuable guidance on how to improve and correct the statistical analyses presented in the manuscript, and provided crucial discussion points that were missing in the first version of the manuscript.

We really appreciate the time and effort Reviewer #1 has put into providing such insightful comments. Indeed, we believe that the manuscript has substantially improved as a result of them. We summarize each of Reviewer's concerns below, and we hope that our responses (preceded by **R:**) are to the Reviewer's satisfaction.

1) *Interpretation of the results*

The authors suggest that species entering and exiting the community tend to be specialists, an inference based on the low relative degree of groups A and C (Fig. 3). However, do these species have a low relative degree because they are simply less abundant? If that's true, then the assembly and disassembly process is simply due to species increasing and decreasing, respectively, in abundance over the course of the season. This would not be surprising and perhaps be a trivial result.

R: We understand the Reviewer's concern and fully agree that species' abundances could be a key factor explaining networks' structural dynamic. Indeed, we are aware of the growing literature exploring this idea, namely the extent to which interaction frequency and variability can be explained by species' abundances, and we regret not having further explored these ideas and discussion points in the first draft of the manuscript. In order to address this, therefore, we have made two major changes to the manuscript.

On the one hand, we used estimates from interaction data and flower counts to study the relationship between species' relative degree and their abundances (L347–353 of the new version of the manuscript; Supplementary Note 2; Supplementary Fig. 14).

While we found a positive relationship between these factors, we also showed that there is considerable variation (85.9% and 66.5% for plants and pollinators, respectively) in species' relative degree that is not explained by abundance. That said, we advise caution when interpreting these results, as pollinators' abundances were not estimated completely independently and instead are inevitably dependent on the observed number of interactions. On the other hand, we added a paragraph to the discussion that explores information on species' abundances as a potential factor explaining the observed structural dynamics (L445–467 of the new version of the manuscript). In hindsight, we realize that this paragraph should have been in the first version of the manuscript, and we are confident that the new additions will be to the Reviewer's satisfaction.

It would help if the authors provide a clear definition of how degree and relative degree were quantified with their weighted network data as well as the sensitivity of these network properties to sampling effects (i.e. abundance).

R: We would like to apologize for the lack of clarity in the definition of relative degree. Following the suggestions provided by Reviewer #1 and #2, we have modified the paragraph where these measures are presented in order to clarify the methodology (L236–241 and L245–250 of the new version of the manuscript). Likewise, we have added sensitivity analyses for species' relative degree, showing that this measure is fairly robust to sampling effects (L241–245 of the new version of the manuscript; Supplementary Fig. 1). In addition, we have also explicitly referenced the sensitivity analyses performed by CaraDonna et al. (2017) in their work introducing the dataset, showing consistent sampling and average detection of interactions for the networks (L117–120 of the new version of the manuscript). We expect that the changes made to the manuscript will clear up the Reviewer's concerns.

In terms of the intermediate dynamics, it appears that there are more patterns to unpack (e.g. common transitions include: stay in A; $B \rightarrow A$ or stay in B; $C \rightarrow A$); however, the biological significance of these "intermediate dynamics" were not discussed much, despite this being framed as one of the novel contributions of this study (Introduction L. 56–58).

R: We agree with the reviewer that we should have explicitly discussed the results regarding species' intermediate dynamics. Please find this discussion in L415–430 of the new version of the manuscript. We thank the Reviewer for this comment as, in hindsight, we realize this again was valuable information that needed to be discussed.

2) *Statistical evidence*

Despite the several assertions made in regard to identifying general assembly rules, there is no statistical support for any of these conclusions. For example, the authors

describe “the position of species entering the network tend to be from group A”, but there is no evidence that this is “above average” or different than another group. They should be able to infer this information from their Bayesian model (which also needs further description, see comments below) and should report this in the Results.

R: We apologize for not reporting the effect sizes for the comparison between transition probabilities in the previous version of the manuscript. Following the Reviewer’s concern, we have now added the effect size for any of the comparisons stated in the Results section (L342–344 of the new version of the manuscript; Supplementary Fig. 11; Supplementary Method 5). We thank the Reviewer for pointing this out.

The authors should also provide statistical evidence supporting their descriptions of groups A, B, and C (I’ve provided one possible way to do this below).

R: We agree that further statistical evidence would solidify our description of the position groups. We followed the Reviewer’s suggestion and used the ‘brms’ R package to do so. Please see the changes made to the Methods (L251–257) and Results (L324–326) sections of the main text, the addition to the Supplementary Method 2, and Supplementary Fig. 3.

3) Bayesian models

The Bayesian models are not described well at all. What priors did the authors use? What statistical software did they use (e.g. they cite McElreath 2018, but its unclear whether they use his “rethinking” package in R)

R: We would like to apologize for the lack of clarity in the description of the Bayesian models. To correct this, we have modified the Methods section of the main text to clearly state the priors and statistical software used (L269–270 and L273–277). Moreover, we have also corrected the description of the models in the Supplementary Information as well as added additional models to test the effect of considering years as temporal replicates (Supplementary Method 4).

We would also like to point out that we decided not to include the density-dependent models to the new version of the manuscript. The reason for this decision is two-fold. First, we realized that such models were qualitatively the same as the time-dependent models. They were all designed to add a temporal component to the transition probabilities, and the only difference was that the density-dependent models used ‘relative number of pollinator/plants species at time $t - \Delta t$ ’ as opposed to ‘relative time’. Following the Reviewers comments, however, we realized that the density-dependent models were much less intuitive and could lead to confusion. Perhaps most importantly, the density-dependent models were not discussed at all throughout the manuscript. In hindsight, we concluded that there was no point in adding an additional source of complexity

to the manuscript if this did not provide any additional ecological insights.

Why don't the authors report any statistical evidence to support their assertions about the assembly rules they claim to have discovered (e.g. effect sizes with credible intervals for each group's transition probability).

R: As mentioned above, we have added the effect sizes for this transition probabilities to the new version of the manuscript (L342–344; Supplementary Fig. 11; Supplementary Method 5). We thank the Reviewer for pointing this out and expect that the changes made to the manuscript will clear up their concerns.

They could also consider using widely applicable information criteria (WAIC) or leave-one-out cross-validation information criteria (LOOIC) to inform the relative support of the different models they analyzed (e.g. transition probabilities are time-dependent, constant, or density-dependent). The authors could then use these criteria to choose which model to highlight in the main text, rather than make what appears to be an arbitrary choice. For example, eye-balling the quantiles in Fig. S9 vs. Fig. 4, makes me think that the constant probability model may be the best choice (smaller quantile range and fewer parameters; in fact, it seems like the authors interpretation of assembly rules follows more from the constant probability model rather than the time-dependent one which is highlighted in the main text..).

R: We fully agree with the Reviewer. In retrospect, we regret not using the WAIC values to compare the different models and assess their performance, in particular because we had already calculated them while exploring the space of possible models. In the new version of the manuscript, we used the model comparison via WAIC and Akaike weights to inform the model selection (L286–291; Supplementary Method 4; Supplementary Fig. 10). The time-dependent model indistinguishably treating the different sampling seasons (the model presented in the main text) had the best performance.

On a related note, why was each year analyzed separately? Why not treat each year as a temporal replicate and test whether the transition probabilities differ between years (which would suggest assembly rules depend on the year, i.e. there are no general rules) or are consistent across years (i.e. there may be general rules for this community).

R: We thank the Reviewer for pointing this out. We fully agree that treating the different years as temporal replicates is a stronger approach to understand whether or not the results that we found are consistent across years. Therefore, we analysed all years together and included the information regarding the sampling seasons to the models in different ways as temporal replicates (a full description of the models can be found in the Supplementary Method 4). Comparing the WAIC values allowed us to more confidently argue that the results were indeed consistent across years, as models using the different

years as temporal replicates did not improve the WAIC scores. We modified the main text to reflect this (L286–291 and L332–336), changed the figures accordingly (Fig. 4; Supplementary Fig. 12–13 and 15), and added an explicit comparison of the models in the Supplementary Information (Supplementary Fig. 10).

4) Identifying node-level properties of each group

If you're comfortable using Bayesian models, one possibility would be to use a multivariate autoregressive model with the brms package in R. Sample code: `brm(mvbind(species, neighbours-average, neighbours-maximum) ~ year + group + (1|species), autocor = cor_ar(formula = ~ week, p = 1))`. This could provide you with information on the effect sizes and credible intervals for the relationship between group and node properties that also accounts for confounding factors (e.g. year, species ID, temporal autocorrelation within each year, and residual correlation among position properties). This approach would also allow you to use all of the information, rather than separating it among years. Note that this is a rough sketch and you may find a better alternative. The key point is that you should show statistical evidence for the claims you make.

R: We thank the Reviewer for providing us this great suggestion. As already mentioned in a previous answer, we followed the Reviewer's ideas and added the corresponding analyses to the main text (L251–257 and L324–326) as well as to the Supplementary Information (Supplementary Method 2; Supplementary Fig. 3). We are confident that the new additions will be to the Reviewer's satisfaction as they much more clearly support our description of key differences across groups.

5) Static networks

I feel the authors build a bit of a "straw man" argument in the Intro and Discussion regarding theory of "static interaction networks". These networks are only static in regard to per-capita parameters (e.g. intrinsic growth rates, inter- and intraspecific interaction rates, handling time, to use Bastolla et al. 2009 as an example). In contrast, population-level interaction strengths do dynamically change in these models as the abundance of plants and pollinators changes. If the authors showed, for example, that per-capita interaction strengths were dynamically changing over time, then I think there would be more room for critique. As is though, the dynamic nature of the plant-pollinator network analyzed here could simply be due to changes in abundances of interacting populations with constant per-capita parameters, which would be consistent with the assumptions of the "static interaction networks". Perhaps the "density-dependent" transition probabilities would give more insight here; however, the authors haven't provided any details on exactly how density dependence was included in that model.

R: We understand the Reviewer's concerns regarding the idea of whether or not mathe-

mathematical models consider ecological communities as static entities. Since it is true that we are not providing evidence that per-capita interaction strengths are dynamically changing over time, we decided to remove the contentious paragraph from the Discussion of the new version of the manuscript. Likewise, we reworded the Introduction to reflect the Reviewer's comments (L32–35 of the new version of the manuscript). We are confident that the changes to the manuscript will clear up the Reviewer's concerns.

6) *Minor comments*

Intro. L. 37: I would only use “evolve” here if you mean “biological evolution” (I assume you aren't referring to biological evolution though based on the references given). I would simply use “change” to be more clear.

R: We thank the Reviewer for pointing this out. We modified the wording in the Introduction to avoid any confusion (L39 of the new version of the manuscript).

Intro. L. 95: “then comprise”? This word choice seems odd to me.

R: The reviewer may be amused to learn this also mirrors a previous discussion between some of our co-authors. We have nevertheless re-checked the Oxford English Dictionary and verified that the word “comprise” is appropriately used in this fashion and has the intended meaning.

Methods (general): For the most part, I think the authors do a good job of clearly describing the “network alignment” methodology, which will be unfamiliar to most ecologists. For example, I thought the ‘Measuring the change of position of individual species’ section was quite clear and had a good example (L. 178–183). Still, there were other places where I needed more help. For example, I got confused in the ‘Position variability across networks’ section. It may be helpful to reference Fig. 1 (which is excellent by the way) at times to help more visual learners understand exactly what your methods are doing. For example, the authors could edit the sentence, “For example, given two networks A_{t_1} and A_{t_2} that were collected at two time points t_1 and t_2 , we wanted to test whether or not any species i present in both networks—i.e. a species $i \in A_{t_1} \cap A_{t_2}$ —changed its position over time (e.g. species a_3 in Fig. 1b).”

R: We thank the Reviewer for pointing this out. We decided to follow their suggestion and modified this part of the Methods section to reference Fig. 1 (see L175, L178, L189 and L191 of the new version of the manuscript).

Methods L. 232–234: “Importantly, species are assumed not to re-enter the network once they have exited it, making the transition from state y_{post} to any other state occur with probability zero”. Although this is a fantastic dataset, my guess is that it is not uncommon for a species to appear to exit the network (e.g. not observed one week), and

then “re-enter”, even though the apparent exit from the network could just be due to sampling error. How was this situation treated? Were these treated as actual “exits” even though they probably were not?

R: The Reviewer is right in assuming that a few species might be missed in one week, appearing to exit the network and re-enter in the following time step. We treat such cases as sampling errors, ignoring both the transitions involved. We had indeed described this in the previous version of the manuscript following the sentence highlighted by the Reviewer, noticing that we treated “species phenologies as uninterrupted”, considering these types of transitions “to be the results of a sampling error”. That said, we slightly reworded this description in order to better clarify the Reviewer’s concern (L282–283 of the new version of the manuscript).

Methods L. 206–216: Why did the authors focus on the ‘short random walks’ algorithm? Is there is a biological/statistical justification? Or is it because this algorithm produced three groups of plants and pollinators, which just felt like a good number of groups for interpretation...

R: We chose the ‘short random walks’ algorithm because it showed agreement with other community detection methods (via normalized mutual information analysis; Supplementary Table 1), and because it produced a number of groups that facilitated their interpretation in purely ecological terms (notice that other groupings were also considered in Supplementary Fig. 9 and 15). That said, we have added a clarification in order to make this justification explicit in the Discussion section (L437–439 of the new version of the manuscript).

Methods L. 208–211: Please provide a clear description of how you quantified degree and relative degree for this weighted network data.

R: We would like to apologize for the lack of clarity in the description of these metrics. As already mentioned in a previous answer, we have changed the wording of the Methods section to clarify these descriptions (L236–241 and L245–250 of the new version of the manuscript)

Results Fig. 2b and Fig. 2c: Intuitively, I feel like the results for variability (Fig. 2b) and quality (Fig. 2c) should be the same, but apparently they give different insights (e.g. plants appear to exhibit less variability in their positions across networks, but the low quality suggests they exhibit more variability than pollinators). It would be worth clarifying here or in the Methods how these different ways to assess change in species position over time have different biological meanings.

R: We would like to point out that Fig. 2c does not represent “alignment quality” but “change of alignment quality”. That is, it provides a way to assess how strongly the

fixing of specific alignments decrease the overall alignment quality. We changed the wording of the caption for Figure 2 to better clarify this distinction. Regarding the link between ‘alignment variability’ and ‘change of alignment quality’, they characterize two different, albeit related, things. While “alignment variability” quantifies whether or not a species changes positions over time, the “change of alignment quality” quantifies the magnitude of such change. Since we noticed that the way we described these measures in the Methods sections was not clear enough, we have reworded them to avoid any confusion (L200–203 of the new version of the manuscript).

Results L. 267: Why highlight the year 2015? It makes me think there is something special about this year, but it’s just one of the temporal replicates right?

R: Following the Reviewer’s previous suggestion regarding the use of sampling seasons as temporal replicates, we are now examine all years together rather than analysing them separately. That said, we kept Fig. 3 as in the previous version because showing only one year is helpful for visualization purposes. We chose to visualize year 2015 because this is the sampling seasons presenting the highest number of observations. We have clarified this in the caption of the figure. We thank the Reviewer for such insightful feedback, and we expect that the changes made to the manuscript should clear up their concerns.

Results L. 281–286: This appears to be a description of the general assembly rules the authors discovered. I think it would be much more helpful for the reader if the authors describe this in biological terms (e.g. species entering the network tend to be specialists that interact with generalists that were already present (i.e. group A)).

R: We have added a clarification to the Results section following this comment (L338–340 of the new version of the manuscript). Moreover, we now provide an extended description in biological terms in the discussion as a separate paragraph (L415–430 of the new version of the manuscript). While putting this together, we have particularly ensured that we clearly describe all transition probabilities outlined in the Results section.

Discussion L. 328–329: “Noticeably, our observations on the variability of species positions also agree with recent work showcasing constant temporal switching of species’ interactions in other empirical plant-pollinator communities (CaraDonna et al., 2017; Miele et al., 2019)”. One of these studies actually contributed the data to this study, so “other” is a bit misleading.

R: We thank the Reviewer for picking up on this. We have changed the wording accordingly (L392 of the new version of the manuscript).

Reply to Reviewer #2

This is a complex, but extremely well-written paper tackling a very relevant question in community ecology. Despite I am not familiar with the methods used, and those are complex, the paper is easy to follow. Most of my comments try to link the methods with the empirical observations to enhance the story.

Reviewer #2 found the manuscript to address a relevant topic in ecology and provided several valuable comments on aspects regarding the Methods, Results and Discussion sections. We really appreciate the time and effort the Reviewer has put into providing feedback for this manuscript. Their comments were really insightful, and we believe that a stronger manuscript has been produced because of them. We have addressed each of those comments and respond to them below (preceded by R:).

1) Sampling effort

My main concern, as always in network ecology, is about sampling effort, and the possibility of having unobserved links in the dataset. A couple of sentences about sampling coverage and how robust is your analysis to unobserved links may reinforce the conclusions.

R: We thank the Reviewer for pointing this out. In hindsight, we realize that this could have been easily included in the first version of the manuscript. We have added sensitivity analyses for species' relative degree, showing that this measure is fairly robust to sampling effects (L241–245 of the new version of the manuscript; Supplementary Fig. 1). In addition, we have also explicitly referenced the sensitivity analyses performed by CaraDonna et al. (2017) in their work introducing the dataset, showing consistent sampling and average detection of interactions for the networks (L117–120 of the new version of the manuscript). We expect that the changes made to the manuscript will clear up the Reviewer's concerns.

2) Species pairing

A second recommendation would be to give a few more details on how the pairing works. Especially related to (i) how the structural roles are defined, which now is only in the supplementary material and (ii) how the pairing works when you compare matrices of very different sampling sizes. This is something that I still don't fully understand. For example, in the section starting at line 142, "species positions". It is unclear to me if the definition of uniqueness depends on the number of species within a network analyzed? I understand uniqueness is a fraction, which may be biased by the number of potential pairs that can be analyzed, but I am not sure. If unequal richness is an issue, how do you take into account species richness?

R: We thank the Reviewer for this suggestion, we agree that some additional details on the alignment method will be useful to the readers. Therefore, we decided to change multiple parts of the methods to reflect the Reviewer’s comment. In particular, we have added a definition of how the structural roles used by the alignment algorithm are calculated (L132–139 of the new version of the manuscript), and some additional information on how our method works for differently sized networks (L155–160 of the new version of the manuscript). Moreover, we also made sure to reference our previous work, where the alignment algorithm is properly defined and tested (Bramon Mora et al. 2018). Regarding the ‘position uniqueness’ measure, we also made sure to slightly change the wording of the section “Position uniqueness within networks” of the Methods to clarify how the measure is calculated, as we realized that the prior wording could lead to unintended confusion. Position uniqueness is a measure of the number of symmetric positions in a network; it therefore characterizes the networks’ structural redundancy. If a species’ position presents low uniqueness, it implies that other species share the same position within the same network (i.e. the same time point). The position of such a species is structurally redundant. For any given time point (i.e. weekly network), position uniqueness is calculated as the *proportion of alignments* that a species is paired to itself *relative to 100 independent alignments performed* (as opposed to the number of potential pairs).

To explain a bit further, in Bramon Mora et al. 2018 some of us showed that large food webs ($\gtrsim 50$ species) might be more redundant than smaller ones, as these have a greater probability of including trophically identical species as well as species that participate in just a single interaction. However, it is unclear to what extent uniqueness depends on species richness or is instead an intrinsic property of large ecological networks. For example, if a network of 10 species has a central node individually connected to the remaining 9 species, each of these 9 species will present very low position uniqueness. In contrast, if a network of 10 species is connected as two non-symmetric 5 species motifs, all 10 species will have high position uniqueness. This argument can be made for any number of species n . Long story short, we share the Reviewer’s interest on this measure, and we are actually currently working on a separate project that explores these ideas further. For the sake of this current study, nevertheless, all species show similar levels of position uniqueness, and if larger networks were significantly more redundant, we would expect to see lower levels of uniqueness in Fig. 2a. That is, controlling for species richness would not change the observed distribution in Fig. 2a in a way that would change our conclusions. With that being said, and following the Reviewer’s comment, we believe that this is a pertinent comment to make with regards to Fig. 2a; therefore, we added Supplementary Note 1, where the relationship between position uniqueness and species richness is discussed.

3) *Minor comments*

Line 116. “where the weight of all interactions was instead set to 1”. I understand what you mean, but you can explain how you binarize the networks in a clearer way.

R: We thank the Reviewer for pointing this out. In hindsight, we realize that this definition was far from clear, and we have changed the wording of this section of the Methods (L122–126 of the new version of the manuscript).

Line 232–234. This means NO species was sampled e.g. in week 2, not in week 3, and again in week 4? This is surprising.

R: We thank the Reviewer for their comment, and for any confusion from the earlier version. We were simply pointing out that a few *individual* species might be missed in some weeks due to a sampling error—appearing to exit the network and re-enter in a subsequent time step. As noted in L279, we treated “species phenologies as uninterrupted”, considering these types of transitions “to be the results of a sampling error”; therefore, we ignored any of such observations when inferring the probabilities. However, at least one species was sampled for all weeks. We have reworded this section to clarify this (L282–283 of the new version of the manuscript).

Line 245. Given the comparison performed, I expected species to resemble the most to themselves (be paired with). Another potential question may be how close is the next potential pair is, right? In any case, I understand uniqueness is expressed in % and I understand this as the effect sizes of your analysis and I would like to see those written here.

R: We agree with the Reviewer that species will most likely be paired with themselves when aligning a network with itself. Indeed, the aim of the analysis in Fig. 2a is to calculate the extend to which this is true. We have added a clarification in L175 and L178 of the new version of the manuscript, referencing Fig. 1 to help readers understand the measure. Regarding the question of how close the next potential pair is, one could use the alignment matrix M presented in the “Identifying distinct groups of species’ positions” section of the Methods to do so. In the manuscript, we do something along these lines to identify the different groups of positions, as the community detection methods grouped species based on their position similarity at any given week (see L226–235). Finally, we were not sure what exactly the Reviewer meant by ‘the effect sizes of [our] analyses’, and we realize that this could potentially be due to a lack of clarity in the way we described the measure and results. Therefore, and following the changes in the Methods sections described above, we have also added clarifications for Fig. 2 to qualitatively interpret the results presented therein (L297–298 and L306–307 of the new version of the manuscript).

Line 254. I agree that it changes across seasons, but given that weekly networks are most

likely incomplete, I wonder how sensitive is your analysis to those sampling biases. For example, if you compare one network, with the same network jittered, how consistent are the positions? I would love to see that small perturbations to the interaction frequencies do not alter drastically the results.

R: We thank the Reviewer for this comment. With regard to the specific analyses presented in Fig. 2b, we do not expect our results to be substantially affected by changes in species' interaction strengths. While the interaction strengths are useful to untangle the structural symmetries between species' positions, these do not change the overall structure of the motif-role profiles (Bramon Mora et al. 2019, *bioRxiv*). We feel that this could actually be seen already in Supplementary Fig. 2, where the analyses for binary networks show species with similar levels of position variability but reduced uniqueness. That is, interaction strengths added crucial information to break the structural symmetries within networks (i.e. lower position uniqueness due to higher structural redundancy; Supplementary Fig. 2a); however, these did not lower the level of structural variability (i.e. changes in species' interactions go beyond changes of interaction strengths and number of observations; Supplementary Fig. 2b).

Following the comments of both Reviewer #2 and Reviewer #1, however, we have added sensibility analyses to the manuscript in order to understand how the random removal of observations will affect species' relative degrees and the overall number of interactions in the networks (L241–244). We found that the networks studied are quite robust to sampling effects. This is in agreement with the results presented by CaraDonna et al. (2017), which are now also explicitly referenced in L117–120 of the new version of the manuscript. Finally, we have also directed the reader (L159 of the new version of the manuscript) to the tests that we presented for the alignment algorithm in Bramon Mora et al. (2018), where the behaviour of the algorithm following the randomization of links and removal of nodes was studied in greater detail. We are confident that the additional clarifications and analyses will be to the Reviewer's satisfaction.

Line 260. Are plant phonologies larger on average than pollinator phonologies? This may explain the discrepancy.

R: We thank the Reviewer for this suggestion. Plant phenologies are indeed longer on average than pollinator phenologies, which might explain the observed discrepancy. We have added the Reviewer's comment to the main text (L312–313 of the new version of the manuscript).

Line 283. I would suggest refreshing the reader that group A is a specialist-generalist interaction, and so on.

R: We agree that a clarification in the Results section could be useful to the readers, and

we have changed the wording throughout the paragraph accordingly (L338–340 of the new version of the manuscript).

Line 300. I would suggest relating those positions to the nested structure of most networks, or to the concept of core species (generalist-generalist), peripheral species (specialist-generalist) and specialized interactions. Of course, a discussion on how abundance may be responsible for these observed patterns is warranted. One key element which may help a lot more empirical readers to add more natural history to the Discussion. For example, show me an example of a real species and how it changes roles along the season.

R: We thank the Reviewer for these insightful discussion points. In retrospect, we regret not having included them initially in the discussion of the earlier version of the manuscript. To correct this, we have first included a comment on the link between the plant-pollinator dynamics and the nested pattern observed in empirical networks (L412–414 of the new version of the manuscript). Second, we have added a paragraph discussing abundances as potential factors explaining the observed dynamics (L445–467 of the new version of the manuscript). Finally, we have incorporated examples of real species to describe the different transition probabilities that we observe (L415–430 of the new version of the manuscript). We are confident that the new additions will be to the Reviewer’s satisfaction.

Line 348. But how much this is driven by abundance patterns. The first and last individuals of any species are likely under-sampled for its interactions, and hence appear only in a small fraction of their potential interaction niche. Hence, what you are observing may be a side effect of a gaussian phenology-activity distribution and not an emergent property of the networks. I am just playing devil’s advocate here but would be a discussion worth having.

R: The Reviewer’s comment is indeed very appropriate, and we again regret not having at least some similar statement in the earlier manuscript. On the one hand, and despite lacking completely independent measures of pollinator abundances to properly test their influence on the observed patterns, we used estimates from interaction data and flower counts to study the relationship between species’ relative degree and their abundances (L347–353 of the new version of the manuscript; Supplementary Note 2; Supplementary Fig. 14). While we found a positive relationship between these factors, we also showed that there is considerable variation in species’ relative degree that is not explained by abundance (85.9% and 66.5% for plants and pollinators, respectively). We advise caution when interpreting these results though, as pollinators’ abundances were estimated from interaction data, which are inevitably dependent on the observed number of interactions. On the other hand, as already mentioned in the comment above, we added a paragraph to the discussion that explores information on species’ abundances as

a potential factor explaining the observed structural dynamics. We thank the Reviewer for the insightful comments and we expect that this will clear up their concerns.

Figure 4. Effect sizes look minor in several panels. This may be better reflected in text and mention the probabilities of each transition (and associated errors) explicitly.

R: Following the Reviewer's concern, we have now added the effect size for any of the comparisons stated in the Results section (L342–344 of the new version of the manuscript; Supplementary Fig. 11; Supplementary Method 5).

Reviewers' Comments:

Reviewer #1:

Remarks to the Author:

I've now reviewed the authors' revision and response letter. I think the authors have done a solid job of addressing my previous concerns. I especially appreciate the inclusion of statistical details and the expanded treatment of sampling/abundance. As I remember from the previous version, the paper was well written, and it continued to be in this revision.

The only detail that gave me pause was Supplementary Figure 14. If we look at the 'Hill' model, it appears that the majority of pollinator observations fall within the region where there is a strong positive relationship with 'relative degree' (e.g., less than 10 observations). At the end of the day though, I thought the authors did a nice job of addressing this issue in their new discussion paragraph when they talked about the "two ends of an interesting conceptual spectrum". This isn't so much of a major comment to be addressed, but I just want to mention that I think this is an unsolvable concern for this manuscript, and I appreciate that the authors have devoted a nice paragraph in their discussion to this issue. They have at least addressed my previous concern on this topic.

I only have some minor comments remaining, which I include below. I don't think they are strictly necessary to address, but I think they could be helpful in clarifying some details for other readers:

L244--245: what does "loss of interactions" exactly mean? I assume it means qualitative interactions, as there is necessarily a loss of interactions with removing observations, right?

L245--250: I was initially lost here. I think the reason was that the verbal description was immediately followed by the mathematical description, but without any distinction of the two. Perhaps just a comma to separate the two, or putting the math in parentheses (i.e., ...) would make this more clear.

L309: 'effect' instead of 'affect'?

L338-344: specialists pollinators interacting with generalist plants, right? I think it could be helpful to be explicit here. The same could be done for L362-364.

Legend of Supp. Fig. 3: don't you mean top-left corner?

Legend of Supp. Fig. 10: vii)? Or should it be vi)? Also, this figure is a little difficult to understand. I would consider including less information. I'm also surprised by the very high weight of model iv) despite some models having similar WAIC values (unless I'm misinterpreting the figure).

I detected a couple of minor grammatical errors (e.g., delete 'an' on L345; delete "d" at end of "calculated" on p. 2 of Supplementary Material after description of priors).

Reviewer #2:

Remarks to the Author:

The authors have addressed all my concerns and this version reads really good. Thank you. The only missing detail is that it would be great if the code is provided directly in a code repository or as supplementary material, and not upon request.

Best,
Ignasi Bartomeus

Reply to Reviewer #1

I've now reviewed the authors' revision and response letter. I think the authors have done a solid job of addressing my previous concerns. I especially appreciate the inclusion of statistical details and the expanded treatment of sampling/abundance. As I remember from the previous version, the paper was well written, and it continued to be in this revision.

The only detail that gave me pause was Supplementary Figure 14. If we look at the 'Hill' model, it appears that the majority of pollinator observations fall within the region where there is a strong positive relationship with 'relative degree' (e.g., less than 10 observations). At the end of the day though, I thought the authors did a nice job of addressing this issue in their new discussion paragraph when they talked about the "two ends of an interesting conceptual spectrum". This isn't so much of a major comment to be addressed, but I just want to mention that I think this is an unsolvable concern for this manuscript, and I appreciate that the authors have devoted a nice paragraph in their discussion to this issue. They have at least addressed my previous concern on this topic.

I only have some minor comments remaining, which I include below. I don't think they are strictly necessary to address, but I think they could be helpful in clarifying some details for other readers.

Reviewer #1 considered the changes made to the manuscript to be of their satisfaction and only provided some minor comments. Overall, we really appreciate the time and effort the Reviewer has put into providing such insightful feedback for this manuscript. We truly believe that a stronger manuscript has been produced thanks to both Reviewers' input. We have addressed the few minor comments made by Reviewer #1 and respond to them below (preceded by R:).

1) Minor comments

L244–245: what does "loss of interactions" exactly mean? I assume it means qualitative interactions, as there is necessarily a loss of interactions with removing observations, right?

R: We understand the Reviewer's confusion and fully agree that "qualitative interactions" is the appropriate term here. We have modified the sentence accordingly in the new version of the manuscript.

L245–250: I was initially lost here. I think the reason was that the verbal description was immediately followed by the mathematical description, but without any distinction of the two. Perhaps just a comma to separate the two, or putting the math in parentheses

(i.e.,) would make this more clear.

R: We would like to apologise for the lack of clarity, and agree that a comma before the mathematical description is clearly needed. Therefore, we have changed this section of the Methods following the Reviewer's suggestion.

L309: 'effect' instead of 'affect'?

R: We thank the Reviewer for picking up on this. We have changed the word accordingly.

L338-344: specialists pollinators interacting with generalist plants, right? I think it could be helpful to be explicit here. The same could be done for L362-364.

R: We thank the Reviewer for pointing this out. We modified the wording in this section to avoid any confusion.

Legend of Supp. Fig. 3: don't you mean top-left corner?

R: Again, we thank the Reviewer for picking up on this. We have corrected the legend for this figure as suggested.

Legend of Supp. Fig. 10: vii)? Or should it be vi)? Also, this figure is a little difficult to understand. I would consider including less information. I'm also surprised by the very high weight of model iv) despite some models having similar WAIC values (unless I'm misinterpreting the figure).

R: First and foremost, we would like to apologise for the confusion with the labelling of the models. In hindsight, we realized that this was a mistake due to changes in the model labelling used in previous versions of the figure. We truly thank the Reviewer for picking up on this. We have corrected the figure legend in the new version of the manuscript. Second, we agree that Supplementary Figure 10 was a bit difficult to understand. Therefore, we've decided to follow the Reviewer's suggestion and include less information. Specifically, we have decided to remove the standard error of the difference between each WAIC and the best performing model. Finally, regarding the weight of the models, this is calculated using the standard error of the difference between each WAIC and the best performing model (now removed from the figure). While the WAIC values overlap a lot between models, the uncertainty about the WAIC is correlated between models. Therefore, one cannot decide between models only based on the WAIC values and their uncertainty, as the correlation between these uncertainties also needs to be considered. The Akaike weights account for these correlations, comparing the relative predictive accuracy of the models. In our case, the standard error of the difference between each WAIC and the best performing model showed model iv to be superior to the rest. With that being said, we have added a clarification to the legend of this figure to

reflect this point. We are confident that the changes to the Supplementary Information will clear up the Reviewer's concerns.

I detected a couple of minor grammatical errors (e.g., delete 'an' on L345; delete "d" at end of "calculated" on p. 2 of Supplementary Material after description of priors).

R: We thank the Reviewer for picking up on these grammatical errors. We have corrected them in the new version of the Supplementary Information.